# Gastric Flora in Gastrostomy Fed Children with Neurological Impairment on Antacid Medication

**DOI:** 10.3390/children7100154

**Published:** 2020-09-29

**Authors:** Bradley De Souza, Susan E. Richardson, Eyal Cohen, Sanjay Mahant, Yaron Avitzur, Sarah Carsley, Adam Rapoport

**Affiliations:** 1Department of Paediatrics, Hospital for Sick Children, University of Toronto, Toronto, ON M5G 1X8, Canada; bdesouza@chla.usc.edu (B.D.S.); eyal.cohen@sickkids.ca (E.C.); sanjay.mahant@sickkids.ca (S.M.); 2Division of Microbiology, Hospital for Sick Children, Toronto, ON M5G 1X8, Canada; susan.richardson@sickkids.ca; 3Department of Laboratory Medicine and Pathobiology, University of Toronto, Toronto, ON M5S 1A8, Canada; 4Institute of Health Policy, Management and Evaluation, University of Toronto, Toronto, ON M5T 3M6, Canada; 5Division of Gastroenterology, Hepatology and Nutrition, Hospital for Sick Children, University of Toronto, Toronto, ON M5G 1X8, Canada; yaron.avitzur@sickkids.ca; 6Department of Health Promotion, Chronic Disease and Injury Prevention, Public Health Ontario, Toronto, ON M5G 1M1, Canada; sarah.carsley@oahpp.ca; 7The Dalla Lana School of Public Health, University of Toronto, Toronto, ON M5T 3M7, Canada; 8Department of Family and Community Medicine, University of Toronto, Toronto, ON M5G 1V7, Canada; 9Emily’s House Children’s Hospice, Toronto, ON M4M 0B7, Canada

**Keywords:** neurologic impairment, gastrostomy tube, aspiration pneumonia, gastro-esophageal reflux, antacid medication

## Abstract

This prospective cohort study aimed to: (1) describe types, concentrations and sensitivity profiles of bacteria found in gastric aspirates of neurologically impaired children; (2) compare flora between outpatients and those admitted with aspiration pneumonia; and (3) examine predictors of bacterial colonization. Gastric aspirates from gastrostomy fed, neurologically impaired children on antacid medication were measured for pH and sent for microbiological testing. The outpatient arm included 26 children at their baseline; the inpatient arm included 31 children with a clinical diagnosis of aspiration pneumonia. Descriptive statistics summarized the ecology and resistance patterns of microbial flora. Predictors of total bacterial colonization were explored with linear regression. High concentrations of potentially pathogenic fecal-type bacteria were detected in 50/57 (88%) gastric aspirates. pH was found to be the only predictor of bacterial growth; children with gastric pH ≥ 4 had significantly higher concentrations of aerobic growth, while those with no bacterial growth had a pH < 4. Further studies to evaluate optimal gastric pH, the role of gastric bacteria in causing aspiration pneumonia, and the optimal empiric therapy for aspiration pneumonia are recommended.

## 1. Introduction

Children with neurological impairment often suffer from oropharyngeal incoordination, placing them at increased risk of feeding-related difficulties. Although targeted strategies to address feeding challenges, such as acid suppression medications and the use of enterostomy tubes (i.e., gastrostomy, gastro-jejunostomy), can result in improved nutritional and growth outcomes in neurologically impaired children, they fail to prevent secondary aspiration events, such as aspiration pneumonitis and pneumonia [1,2,3]. Aspiration pneumonitis is an inflammatory process that occurs when an inhaled toxic substance, typically gastric contents, causes acute lung injury; aspiration pneumonia is an infectious process that results when inhaled secretions from above (i.e., oropharynx) or below (i.e., gastric contents) contain a sufficient bacterial load [4]. Although significant overlap can occur with these events, neurologically impaired children with a suspected aspiration event are generally assumed to have pneumonia and empirically started on broad-spectrum antibiotics [5]. As the leading cause of chronic lung disease and most frequent cause of death in this patient population, the prevention of aspiration events is a critically important focus for treatment of this vulnerable population [3,6,7].

Increasingly, acid suppression medications, including histamine-2-receptor antagonists and proton pump inhibitors, are prescribed to children with neurological impairment in an attempt to reduce symptoms and consequences of gastroesophageal reflux disease [8]. Practitioners have also traditionally prescribed these medicines in an attempt to protect patients from developing chemical pneumonitis related to the potential aspiration of caustic gastric contents [4,9]. Recent data, however, have shown that this approach may paradoxically place patients at an increased risk of pneumonia by increasing the probability of developing bacterial lung infections, possibly related to enhanced bacterial growth in the stomach as a result of gastric acid suppression [10,11,12,13].

In an effort to better understand the relationship between gastric acid suppression in neurologically impaired children and the potential for bacterial colonization, this study aimed to: (1) describe the type, concentration and sensitivity profile of bacteria found in gastric aspirates of this population; (2) compare the gastric microbial flora among inpatients admitted with a clinical diagnosis of aspiration pneumonia to an outpatient group with similar characteristics, but no clinical aspiration pneumonia; and (3) to examine predictors of bacterial colonization of the stomach, including types and degree of acid suppression, the method of gastric feeding (continuous versus bolus feeds) and recent antibiotic use.

## 2. Patients and Methods

### 2.1. Design and Setting

This prospective cohort study involved children (<18 years of age) with neurological impairment who were fed by gastrostomy tube (G-tube) and who received regular acid suppression medication (e.g., sucralfate, histamine-2-receptor antagonists, proton pump inhibitor). The study protocol was approved by the research ethics board at The Hospital for Sick Children, Toronto, Canada (REB File No.: 1000028115). Informed consent was obtained from the parents of all subjects.

### 2.2. Sample

Outpatients were eligible if they were otherwise well on a routine visit; inpatients had to be admitted with a clinical diagnosis of aspiration pneumonia. The diagnosis of aspiration pneumonia was determined by the responsible physician and based on clinical and radiologic findings consistent with aspiration pneumonia (i.e., rapid-onset respiratory distress, usually preceded by a coughing/choking spell during a feed or after vomiting, fever, an increase in oral or pulmonary secretions or new chest X-ray infiltrates), in accordance with current best clinical practice [14]. Immunocompromised patients, as well as those on home ventilation [e.g., patients with a tracheostomy, Biphasic Positive Airway Pressure (BiPAP), Continuous Positive Airway Pressure (CPAP)] were excluded due to the potential for these situations to impact the commensal flora and alter the risk of infection, including aspiration pneumonia. Patients with enterostomy tubes not directed into the stomach (e.g., gastro-jejunostomy tubes) were also excluded owing to our inability to easily obtain a sample from the stomach.

### 2.3. Data Collection

Baseline data were collected for all enrolled children including: demographics (age, gender, underlying diagnosis); use of systemic antibiotics for greater than 48 h within two weeks prior to recruitment and reason for antibiotic use; date of G-tube insertion; method of feeding (bolus versus continuous); number of pneumonia episodes in the previous year; class of acid suppression medications (sucralfate, histamine-2-receptor antagonists, proton pump inhibitor), the dose (mg/kg/day) and the duration of use.

All gastric aspirates (minimum of 0.5 mL) were collected from G-tubes at the time of recruitment using a sterile syringe. Inpatient study samples were collected within 2 h of antibiotic administration for the treatment of aspiration pneumonia. G-tube openings were cleansed with alcohol prior to aspirate collection into a sterile container to minimize the risk of contamination. The pH of each specimen was tested using pH strips (BDH^®^ pH Test Strip; pH range 2.0–9.0) and then sent to the Microbiology Laboratory for testing. An anaerobic swab (BBL™ Vacutainer Anaerobic Specimen Collector, Becton Dickinson, Baltimore, MD, USA) of the fluid was also submitted for culture. Fluid was subjected to semi-quantitative aerobic culture (in-house developed protocol) to determine the identification and concentration of each aerobic and facultative anaerobic bacterium present (aerobes plus facultative anaerobes are termed “aerobes” for the duration of the manuscript, to differentiate them from strict anaerobes). Three 10-fold dilutions of each specimen were prepared in sterile saline. A 10 µL aliquot of each of the neat and diluted specimens was inoculated onto 5% sheep blood agar and Brucella blood agar (BBA, Becton Dickinson, Baltimore, MD, USA) plates, and incubated aerobically in CO_2_ for 48 h. The neat specimen alone was inoculated onto bile salt agar and incubated aerobically for 48 h. The anaerobic swab was inoculated onto BBA, phenylethyl alcohol and gentamicin kanamycin agars and incubated anaerobically for 5 days. Each colony type had a Gram stain and basic identification tests performed to permit identification to at least genus level if possible. Bacterial identification was performed according to standard microbiologic practice [15]. Bacterial concentration in colony forming units (CFU) per mL was determined for each aerobe. Total aerobic colony count (TACC) was determined per specimen. Susceptibility testing was carried out on clinically significant aerobic pathogens only, such as *Staphylococcus aureus, Enterococcus* species and enteric gram-negative bacilli. Susceptibility testing was carried out by using Kirby-Bauer disk susceptibility testing for *S. aureus* and *Enterococcus* species, and by the BD Phoenix^TM^ Automated Identification and Susceptibility Testing System (Becton Dickinson, Baltimore, MD, USA) using the Gram Negative MIC panel. Antimicrobial susceptibility testing and interpretation were performed in accordance with the accepted standards [15].

Bacteria were classified into 3 descriptive categories: (1) Fecal-type flora [Enterobacteriaceae (ENT), e.g., *Escherichia coli*, *Klebsiella* species; ampC beta-lactamase-producing gram negative bacteria (SPICE), i.e., *Serratia*, *Providencia*, indole-positive *Proteus*, *Citrobacter*, *Enterobacter* species; Non-fermentative gram negative bacilli (NFB), e.g., *Pseudomonas aeruginosa*, *Acinetobacter*, *Stenotrophomonas*, *Comamonas*, *Chryseobacterium* species; *Enterococcus* species (ENC); Gram positive bacilli (GPB), e.g., *Lactobacillus* species]; (2) Oral flora (viridans *Streptococcus*, *Streptococcus pneumoniae*, *Moraxella catarrhalis*, *Rothia mucilaginosa*, *Neisseria* species, yeast); and (3) Skin flora (*Staphylococcus aureus*, coagulase negative *Staphylococcus*, diphtheroids, *Bacillus* species).

### 2.4. Data Analysis

Descriptive statistics (means with standard deviations, medians with interquartile ranges, proportions with 95% confidence intervals) were performed to summarize the ecology of the microbial flora, the number of children colonized, the type and amount of bacteria and the bacteria resistance patterns. Independent t-test, chi-squared statistic, Fisher’s exact test and the Wilcoxon Rank Sum were used, where appropriate, to assess the differences between children attending clinics (outpatients) and those admitted with aspiration pneumonia (inpatients). Univariate analyses examined the relationship between all independent variables and TACC. The primary outcome of TACC was not normally distributed therefore a log-linear transformation was performed. An adjusted linear regression was performed to explore predictors of total bacterial colonization. Variables entered into the model included age, gender, gastric pH, continuous versus bolus feeds, inpatient versus outpatient status and recent use of antibiotics for more than 48 h in the previous two weeks prior to recruitment. The criterion for statistical significance (*p*-value) was set at two-sided alpha equals to 0.05.

## 3. Results

During the study period, 78 subjects were enrolled; 35 outpatients (no aspiration pneumonia) and 43 inpatients (admitted with aspiration pneumonia). Nine outpatient subjects were excluded due to ineligibility [tracheostomy (*n* = 1); no longer being on acid suppression medication (*n* = 1); gastro-jejunostomy tube in situ (*n* = 1); or an inability to collect a sufficient (≥0.5 mL) gastric sample (*n* = 6)]. A total of 12 inpatients were excluded [no longer on acid suppression medication (*n* = 1); inability to collect a sufficient (≥0.5 mL) gastric sample (*n* = 11)]. Fifty-seven subjects remained: 26 outpatients and 31 inpatients.

The two groups were similar with respect to age, gender, type of acid suppression medication and gastro-enterostomy tube, method of feeding and administration of a prokinetic agent [Table 1]. Median total bacterial counts and mean gastric pH of gastric samples were also comparable between inpatient and outpatient groups. Outpatients, however, were more commonly fed orally (54% vs. 16%, *p* = 0.003), while inpatient subjects were more likely to have received antibiotics within the two weeks prior to study sample collection (45% vs. 15%, *p* = 0.02). Additionally, underlying diagnoses leading to neurological impairment differed between inpatient and outpatient groups although these differences did not meet our statistically significant threshold. For example, underlying genetic syndromes were more prevalent in the outpatient group (50% vs. 26%, *p* = 0.18), while the inpatient group had more participants suffering from cerebral palsy unrelated to genetic syndromes (55% vs. 27%, *p* = 0.18).

Multiple types of bacteria were isolated from gastric fluid cultures. Aerobic bacteria were isolated from 88% of all patients. Anaerobes were isolated in 11 patients (19%), 5 outpatients and 6 inpatients, and always in addition to aerobic bacteria. Although quantitative anaerobic culture was not performed, heavy anaerobic growth was noted in all 5 outpatients, while among the 6 inpatients, 2 had heavy growth, 1 had moderate growth and 3 had light growth. TACC in inpatients and outpatients combined ranged from 3 × 10^2^ to 1 × 10^9^ CFU/mL. The median TACC was not significantly different between inpatients (8.1 × 10^5^ CFU/mL) and outpatients (6.8 × 10^6^ CFU/mL) (*p* = 0.12).

Fecal-type bacteria (ENT, SPICE, NFB, ENC, GPB), rather than flora representative of the oropharyngeal tract or skin, were responsible for 96.7% of the TACC in inpatient gastric aspirates. In contrast, fecal-type bacteria made up only 57.6% of the TACC in outpatient gastric aspirates, while oral and skin type bacteria (including yeasts) were much more common as a proportion of the ACC (42.4%) [Figure 1].

Yeast-type organisms were isolated in 12 patients (21%), 5 outpatients and 7 inpatients. Total combined yeast counts ranged from 1 × 10^2^ to 4 × 10^6^ CFU/mL, with a median of 3.1 × 10^3^ and 2.3 × 10^4^ CFU/mL in outpatients and inpatients, respectively. Seven patients (12%), 4 outpatients and 3 inpatients, had no bacterial growth from their gastric aspirates, with all but one patient having a mean/median gastric pH of 2 [Figure 2]. All patients with no bacterial growth had a pH < 4, while every child with gastric pH ≥ 4 had some bacterial growth.

In multivariable analyses, a one unit increase in gastric pH was found to be independently associated with a more than a 7-fold increase in TACC [RR 7.57 (95% CI 4.18, 13.7)] when adjusting for age, gender, method of feeding, inpatient versus outpatient status and recent antibiotic use [Table 2]. Covariates in the model were not found to be independently predictive of total bacterial growth.

Table 3 shows antibiotic susceptibility patterns for organisms considered to be clinically significant, which did not include anaerobes, most oral and skin type bacteria, most *Enterococcus* species or gram-positive bacilli (mainly *Lactobacillus* spp.). All *Enterococcus* species were found to be sensitive to ampicillin, though broader antimicrobial susceptibility for ENC species was not performed. Due to the relatively small number of isolates of individual species of organisms tested, data were combined for an antibiogram according to categories for which similar antibiotics were reported (all gram-negative bacilli, Enterobacteriaceae including SPICE bacteria, non-fermenting gram negative bacilli). It was not possible to evaluate the antibiograms of inpatients versus outpatients due to the low number of isolates, but there was no observable difference between the two with respect to antibiotic susceptibility.

## 4. Discussion

This study examines bacterial growth in the gastric aspirates of a group of patients prone to aspiration pneumonia—neurologically impaired children fed via a gastrostomy tube and treated with acid suppression medication. Although traditionally considered a sterile environment as a result of bacterial growth suppression by the normally acidic pH of the stomach (pH ≤ 2) [16], this is the first study that we are aware of to demonstrate that a variety of microorganisms can be found in the gastric aspirates of neurologically impaired children. The majority of patients in this study were found to be colonized with high concentrations (mean 2.6 × 10^7^ CFU/mL) of potentially pathogenic fecal-type bacteria (including anaerobes), while a smaller proportion were found to also be colonized with bacteria found in the oropharynx and on the skin. This is, in some important ways, quite different from previous studies reporting gastric aspirates obtained from children taking acid suppression medication evaluated for chronic cough who were predominantly found to have organisms representative of skin and the oropharyngeal tract [17]. The findings also differ from reports of healthy adult males on cimetidine, who had a relatively low mean concentration of mostly oral flora (3.2 × 10^4^ CFU/mL) and in whom gram negative bacilli were distinctly unusual [18]. Patients in both of these previous studies are at substantially reduced risk of aspiration pneumonia than those who are neurologically impaired.

It is notable that pH was a strong independent predictor of bacterial colonization of the stomach in our cohort (whether the member of the cohort was an in- or out-patient). The more successful the acid suppressive medication, the higher the pH and the bacterial load. North American and European guidelines for the treatment of gastroesophageal reflux in children identified proton pump inhibitors as being ‘superior’ to histamine-2 receptor antagonists because of their ability to maintain the gastric pH at or above 4 [8]. Interestingly, we found that a pH ≥ 4 was a predictor for bacterial growth, while all the gastric aspirates among children that failed to grow bacteria had a pH of less than 4.

Few statistically significant differences were found between the inpatient and outpatient arms of our study, despite a clinical diagnosis of aspiration pneumonia in the former. Inpatients with aspiration pneumonia were more likely to have received antibiotics in the two weeks prior to sample collection than were outpatients. It is possible that the use of these antibiotics may have contributed to the aspiration pneumonia, as vomiting can be an adverse effect associated with many antibiotics. However, we hypothesize that many antibiotic prescriptions in this group reflected an attempt to treat early signs of a respiratory tract infection in this susceptible cohort. A greater number of patients in the outpatient arm of the study received some of their nutrition orally. Although this did not seem to impact the presence or concentration of bacteria in their gastric aspirates, it is possible that this may have influenced the higher rates of oropharyngeal flora cultured from outpatient samples.

With the knowledge that bacterial growth in this population may be directly related to increases in gastric pH secondary to acid suppression medications, re-evaluation of current best practice guidelines for treating gastroesophageal reflux disease in children with neurological impairment may be warranted. Prescribing acid suppression medications has been shown to place children at higher risk of community acquired pneumonia [11,12,13]. In light of the high rates and concentrations of gastric bacteria found in our study, it is conceivable that efforts to ameliorate the effects of gastroesophageal reflux disease in neurologically impaired children with acid suppressors may inadvertently put them at greater risk of bacterial pneumonia secondary to refluxed gastric contents, a normally sterile site. As such, a more prudent approach may be to provide antacids for a limited amount of time and to regularly re-assess symptomatology for the clinical necessity of continuing patients on an acid suppression regimen, as recommended in the latest international guidelines [19].

While no correlation has been made between the presence of gastric bacteria and the infectious pathogen(s) implicated in aspiration pneumonia, it is tempting to consider the susceptibility findings from our study and how they might influence the choice of antibiotics in clinical practice. Empiric antibiotics are recommended for treating aspiration pneumonia in neurologically impaired patients, such as those in our cohort, and although coverage should be broad, anaerobic activity is not usually recommended [4,20]. Although we were not able to quantify anaerobic growth in this study, anaerobes were isolated in a minority of patients (19%), and were adequately covered by the usual antimicrobial regimens used (e.g., cefotaxime, ciprofloxacin, piperacillin-tazobactam). Appropriate coverage of gram-negative bacilli was a concern in our patients, but we did not observe high rates of resistance. Although non-fermenting gram negative bacilli (e.g., *Pseudomonas aeruginosa*) are more resistant to antibiotics, especially cefotaxime, which is commonly used in empiric treatment in these patients, they represented only 1–2% of the total bacterial load in both in- and out-patient cohorts, and probably do not need routine targeted empiric coverage. Thus, if bacteria found in gastric aspirates could serve as a proxy for the targeted pathogens in aspiration pneumonia, a single antibiotic such as piperacillin-tazobactam, ciprofloxacin or cefotaxime should be adequate empiric therapy in this setting to cover anaerobes, without additional anaerobic coverage such as clindamycin. In fact, in our inpatients with pneumonia, gram negative bacilli (ENT, SPICE and NFB combined) made up only 6% of the total bacterial load isolated. The organisms detected in highest concentration, such as *Enterococcus* spp., *Lactobacillus* spp. and many oral bacteria are not well-proven agents of pneumonia and are susceptible to narrower spectrum agents like penicillin, ampicillin, amoxicillin-clavulanic acid or clindamycin alone. It is not surprising that oral type bacteria represented a significantly lower proportion of the bacterial load in inpatients with pneumonia compared to outpatients, as they were more likely to have been exposed to antibiotics in the two weeks prior to admission. Antibiotics would have significantly reduced the more susceptible oral type bacterial flora and replaced it with fecal type flora, requiring broader antimicrobial coverage. Skin flora comprised a minority of the bacterial flora in both inpatients and outpatients. Susceptibility testing was only performed on *S. aureus* isolates in this group, and none of these were found to be methicillin resistant; thus skin flora had little impact on antibiotic choices.

A significant limitation of our study is the implication that gastric aspirates may be used as a proxy to provide information about the bacterial cause and treatment of aspiration pneumonia. This hypothesis remains unproven and further elucidation of the role and concentration of specific bacteria in the etiology of aspiration pneumonia in these children is needed. Studies correlating organisms which colonize the stomach with those causing bacterial aspiration pneumonias, through bronchoalveolar lavages, would yield valuable information to further support clinical decision making of physicians treating this vulnerable patient population. Furthermore, in orally fed patients’ pneumonias may result from aspiration during the swallowing of food or oral secretions. The presence and impact of gastric bacteria in patients who aspirate “from above” is unknown.

## 5. Conclusions

Although the prevailing belief is that that stomach contents are sterile as a result of the naturally acidic environment, neurologically impaired children taking acid suppression medication and fed via gastrostomy were found to have high concentrations of bacteria in the stomach, primarily fecal-type bacteria. Increases in gastric pH were directly correlated with increases in total gastric aerobic colony counts. Acquired antimicrobial resistance was uncommon despite the antibiotic pressure to which many of these children are subjected on a regular basis. Assuming a pathogenic role of bacteria aspirated into the lungs from the stomach in children with neurologic impairment on acid suppressive medication, our study suggests that it may be possible to employ more targeted, narrower spectrum empiric antimicrobial therapy such as penicillin, ampicillin, amoxicillin-clavulanic acid or clindamycin for this clinical indication.

## Figures and Tables

**Figure 1 children-07-00154-f001:**
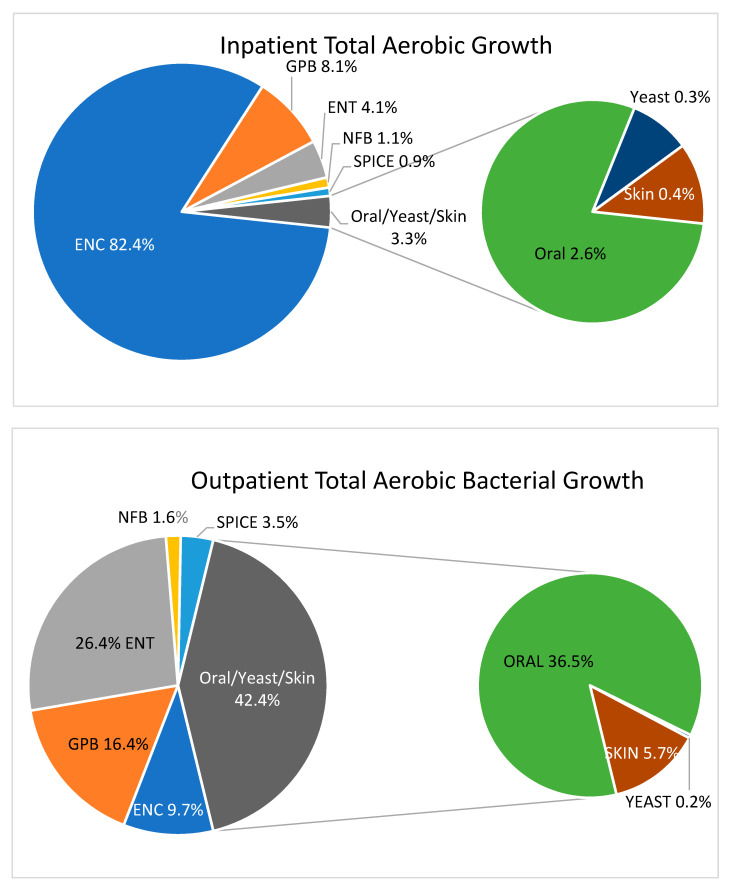
Percent total aerobic bacterial growth by bacterial category. Non-Fermenting Gram Negative Bacilli (NFB)—*Pseudomonas aeruginosa*, *Acinetobacter* spp., *Chryseobacterium* spp., *Stenotrophomonas* spp. Oral Flora—viridans group *Streptococcus*, *Rhodotorula mucilaginosa*, *Moraxella* spp., *Neisseria* spp., *Yeast* spp. *Enterococcus* species (ENC) Enterobacteriaceae (ENT)—*Escherichia coli*, *Klebsiella* spp. SPICE Bacteria—*Serratia* spp., *Providencia* spp., *indole*-*positive Proteus* spp., *Citrobacter* spp. *Enterobacter* spp. Skin Flora—*Staphylococcus aureus*, coagulase negative *Staphylococcus*, *Bacillus* spp., *Corynebacterium* spp. Gram Positive Bacilli (GPB)—*Lactobacillus* pp.

**Figure 2 children-07-00154-f002:**
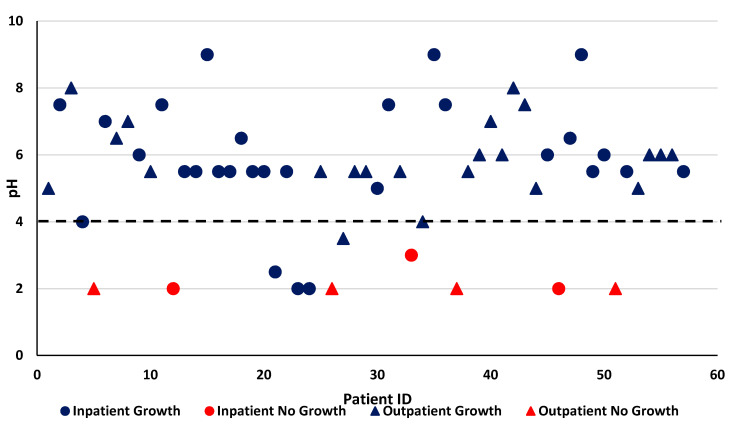
Presence or Absence of Bacterial Growth by pH.

**Table 1 children-07-00154-t001:** Baseline Clinical Demographics.

	Total	Outpatient	Inpatient	*p*-Value
	(N = 57)	(N = 26)	(N = 31)	
Age in Months, Median (Interquartile Range)	37 (25–74)	40.5 (25.5–65.5)	37 (20–86.5)	0.64
Male, N (%)	29 (51)	10 (39)	19 (61)	0.09
Diagnosis, N (%)				0.18
Genetic Syndrome	21 (37)	13 (50)	8 (26)	
Cerebral Palsy	24 (42)	7 (27)	17 (55)	
Metabolic Disorder	6 (11)	3 (12)	3 (10)	
Other	6 (11)	3 (12)	3 (10)	
Orally Fed, N (%)	19 (33)	14 (54)	5 (16)	0.003
Continuous Feeds, N (%)	38 (68)	17 (65)	21 (70)	0.71
Gastrostomy Tube Type *, N (%)				
Mic-Key	13 (23)	5 (19)	8 (26)	0.56
Cook	44 (77)	21 (81)	23 (74)	
Acid Suppression Medication, N (%)				
Omeprazole	50 (88)	22 (85)	28 (90)	0.51
Ranitidine	2 (4)	0 (0)	2 (7)	0.5
Lansoprazole	3 (5)	2 (8)	1 (3)	0.59
Rabeprazole	1 (2)	1 (4)	0 (0)	0.46
Other	1 (2)	1 (4)	0 (0)	0.46
Prokinetic Medication, N (%)	35 (61)	17 (65)	18 (58)	0.57
Received Antibiotics in Two Weeks Prior To Sample Collection, N (%)	18 (32)	4 (15)	14 (45)	0.02
Mean Gastric pH (Standard Deviation)	5.5 (2)	5.4 (2)	5.6 (2)	0.7
pH ≤ 4, N (%)	13 (23)	6 (23)	7 (23)	0.96
pH > 4, N (%)	44 (77)	20 (77)	24 (77)	0.96
Colony Forming Units/mL, Median (Interquartile Range)		6.8 × 10^6^ (8.5 × 10^5^–1.5 × 10^7^)	8.1 × 10^5^ (4 × 10^4^–1.3 × 10^7^)	0.12

* Note: Mic-Key type gastrostomy tubes are low profile tubes, while Cook type gastrostomy tubes are non-low profile tubes.

**Table 2 children-07-00154-t002:** Predictors of Total Bacterial Colonization.

Predictor	Beta Coefficient (β)	Adjusted RR, e^β^ (95% Confidence Interval)	*p*-Value
Omnibus F-test			<0.01
Child Age	0.002	1.0 (0.9, 1.1)	0.9
Sex (Female vs. Male)	0.05	1.1 (0.09, 12)	0.9
pH	2.0	7.6 (4.2, 13.7)	<0.01
Tube Fed (No vs. Yes)	0.3	1.4 (0.08, 24.8)	0.8
Inpatient vs. Outpatient	−1.4	0.3 (0.02, 2.7)	0.3
Recent Use of Antibiotics (No vs. Yes)	−0.9	0.4 (0.03, 4.9)	0.5

**Table 3 children-07-00154-t003:** Inpatient and Outpatient Susceptibility Patterns (%).

	N	Piperacillin/Tazobactam	Cefotaxime or Ceftriaxone	Ceftazidime	Ciprofloxacin	Gentamicin	Tobramycin	Amikacin	Meropenem
All GNB	87	95	44	94	99	90	90	93	95
ENT + SPICE	57	97	95	97	98	93	91	97	100
NFB	30	93	3	90	100	83	87	87	87

*Enterococcus* species, which are not typically considered primary pathogens on their own, had 100% sensitivity to Ampicillin. All isolates of *Staphylococcus aureus* were uniformly susceptible to oxacillin (i.e., methicillin susceptible). Gram Negative Bacilli (GNB)—(ENT, SPICE and NFB combined). Enterobacteriaceae (ENT)—*Escherichia coli, Klebsiella* spp.SPICE Bacteria—*Serratia* spp., *Providencia* spp., *indole-positive Proteus* spp., *Citrobacter* spp. *Enterobacter* spp. Non-Fermenting Gram Negative bacilli (NFB)—*Pseudomonas aeruginosa, Acinetobacter* spp., *Chryseobacterium* spp., *Stenotrophomonas* spp.

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
