# Peer review of "Gastric Flora in Gastrostomy Fed Children with Neurological Impairment on Antacid Medication"

_children, 2020, doi:10.3390/children7100154_

Round 1

Reviewer 1 Report

Line 81 Change: "gastrojejunostomy" in "gastro-jejunostomy"

In table 1 the Authors talk about the use of prokinetics in some patients,
since antibiotics such as prokinetics are sometimes used,
it is important to clarify which patients have used prokinetics
and what type of prokinetics.

Author Response

Authors' reply to Reviewer 1:

Thank you very much for your positive review of our manuscript and for your helpful suggestions. Below you will find an itemized response to your comments. In addition, I have attached a revised version of our manuscript which includes our edits in Track Changes format.

  1. Line 81 Change: "gastrojejunostomy" in "gastro-jejunostomy" - gastrojejunostomy has now been changed to gastro-jejunostomy in line 88 (the line number has changed from 81 to 88 as a result of other edits).
  2. In table 1 the Authors talk about the use of prokinetics in some patients, since antibiotics such as prokinetics are sometimes used, 
    it is important to clarify which patients have used prokinetics 
    and what type of prokinetics. Reviewer 1 raises a good point. Unfortunately, the specific type of prokinetic patients were using was not recorded during our data collection and we are unable to go back to our subjects to collect this missing data. However, while some prokinetics do have strong antibiotic properties (ie. erythromycin), the practice in our institution and geographic region is to not use this medications. Rather, all patients enrolled in our study would have been taking one of the following prokinetics only: domperidone, metoclopramide or cisapride. All 3 of these prokinetics are believed to have minimal if any antimicrobial effect. We have elected to leave the manuscript as is, but should the Reviewer feel that something regarding this must be included, we would be happy to address this in the study limitations paragraph of our Discussion.

Reviewer 2 Report

Thank you for the opportunity to review this manuscript. In this study, the authors set out to describe types, concentrations and sensitivity profiles of bacteria found in gastric aspirates of neurologically impaired children, draw comparisons between the flora of outpatients and inpatients admitted with aspiration pneumonia, and explore predictors of bacterial colonisation. A main finding was that pH was the only predictor of bacterial growth – specifically, children with no bacterial growth had a pH < 4. The authors conclude that gastric pH, as it pertains to chest infection, may need to be carefully considered in this population. I found this manuscript to be exceptionally well-written, with sound rationales and methods presented, and thoughtful interpretation of findings. I believe this work would make a strong contribution to the literature in this field. I only have three minor suggestions that I believe will further strengthen this work:

  1. Please carefully define and consistently use/distinguish the terms aspiration pneumonia and aspiration pneumonitis throughout the manuscript. These terms have very different pathogeneses and (very likely) distinct bacterial profiles, and it was, at times, hard to determine which term/process the authors were referring to.
  2. Please provide rationales for your exclusion criteria.
  3. Please reference your protocols for culturing and quantifying bacteria

Author Response

Authors' reply to Reviewer 2:

Thank you very much for your positive review of our manuscript and for your helpful suggestions. Below you will find an itemized response to your comments. In addition, I have attached a revised version of our manuscript which includes our edits in Track Changes format.

  1. Please carefully define and consistently use/distinguish the terms aspiration pneumonia and aspiration pneumonitis throughout the manuscript. These terms have very different pathogeneses and (very likely) distinct bacterial profiles, and it was, at times, hard to determine which term/process the authors were referring to. We are grateful to Reviewer 2 for highlighting this important distinction. The following definition has now been included in the introductory paragraph (Lines 44-50): Aspiration pneumonitis is an inflammatory process that occurs when an inhaled toxic substance, typically gastric contents, causes acute lung injury; aspiration pneumonia is an infectious process that results when inhaled secretions from above (i.e. oropharynx) or below (i.e. gastric contents) contain a sufficient bacterial load [4]. Although significant overlap can occur with these events, neurologically impaired children with a suspected aspiration event are generally assumed to have pneumonia and empirically started on broad-spectrum antibiotics [5]. In addition, we have reviewed the entire manuscript for instances of these 2 terms and ensured that appropriate term was used.
  2. Please provide rationales for your exclusion criteria. The following rationales have now been provided (Lines 84-89): Immunocompromised patients, as well as those on home ventilation [e.g. patients with a tracheostomy, Biphasic Positive Airway Pressure (BiPAP), Continuous Positive Airway Pressure (CPAP)] were excluded due to the potential for these situations to impact the commensal flora and alter the risk of infection, including aspiration pneumonia. Patients with enterostomy tubes not directed into the stomach (e.g. gastro-jejunostomy tubes) were also excluded owing to our inability to easily obtain a sample from the stomach.
  3. Please reference your protocols for culturing and quantifying bacteria. Our manuscript now states that our culturing methods utilized an in-house developed protocol (Lines 103-107). With respect to bacterial identification techniques and susceptibility testing, we now cite standard microbiologic practice and protocols described by the Clinical and Laboratory Standards Institute (reference #15) and state the following (Lines 113-122):

    Bacterial identification was performed according to standard microbiologic practice [15]. Bacterial concentration in colony forming units (CFU) per mL was determined for each aerobe. Total aerobic colony count (TACC) was determined per specimen. Susceptibility testing was carried out on clinically significant aerobic pathogens only, such as Staphylococcus aureus, Enterococcus species and enteric gram-negative bacilli. Susceptibility testing was carried out by Kirby-Bauer disk susceptibility testing for S. aureus and Enterococcus species, and by the BD PhoenixTM Automated Identification and Susceptibility Testing System (Becton Dickinson, Baltimore, USA) using the Gram Negative MIC panel. Antimicrobial susceptibility testing and interpretation were performed in accordance with the accepted standards [15].
